# Biofilm Formation and Control of Foodborne Pathogenic Bacteria

**DOI:** 10.3390/molecules28062432

**Published:** 2023-03-07

**Authors:** Xiaoli Liu, Huaiying Yao, Xihong Zhao, Chaorong Ge

**Affiliations:** Research Center for Environmental Ecology and Engineering, School of Environmental Ecology and Biological Engineering, Wuhan Institute of Technology, Wuhan 430205, China

**Keywords:** foodborne pathogens, food safety, biofilm, resistance, quorum-sensing

## Abstract

Biofilms are microbial aggregation membranes that are formed when microorganisms attach to the surfaces of living or nonliving things. Importantly, biofilm properties provide microorganisms with protection against environmental pressures and enhance their resistance to antimicrobial agents, contributing to microbial persistence and toxicity. Thus, bacterial biofilm formation is part of the bacterial survival mechanism. However, if foodborne pathogens form biofilms, the risk of foodborne disease infections can be greatly exacerbated, which can cause major public health risks and lead to adverse economic consequences. Therefore, research on biofilms and their removal strategies are very important in the food industry. Food waste due to spoilage within the food industry remains a global challenge to environmental sustainability and the security of food supplies. This review describes bacterial biofilm formation, elaborates on the problem associated with biofilms in the food industry, enumerates several kinds of common foodborne pathogens in biofilms, summarizes the current strategies used to eliminate or control harmful bacterial biofilm formation, introduces the current and emerging control strategies, and emphasizes future development prospects with respect to bacterial biofilms.

## 1. Introduction

Biofilms are defined as communities of microorganisms that are attached to living or abiotic surfaces, and they are common to the growth patterns of microorganisms in nature. Biofilms offer resistance to extreme environments and can protect microorganisms from ultraviolet (UV) radiation, extreme pH, extreme temperature, high salinity, high pressure, malnutrition, antibiotics, etc., thus acting as “protective clothing” for microorganisms [1]. The resistance of biofilms to environmental extremes allows for the creation of suitable habitats for microbial populations and facilitates material and information exchange between microorganisms; thus, biofilms are self-protective mechanisms in microbial growth [2]. The morphological structure, sensitivity to environmental factors, and biological characteristics of microorganisms in biofilms are quite different from those of plankton, and the three-dimensional structure of biofilms also appears to provide a natural barrier and protective layer for microorganisms [3]. In addition, studies have now established that biofilms are the primary source of contamination during food contamination and that the persistence of biofilms on food contact surfaces and equipment is a key factor that serves as an enduring source of contamination.

In the food industry, some microorganisms that are inherent in food products are harmless to consumers and in some cases may provide some benefits (for example, microorganisms introduced in the form of ferments in fermented foods, probiotics in yogurt, Saccharomyces cerevisiae in rice wine, etc.). Therefore, unless there is excessive growth or visible food spoilage, no effort is usually made to remove such microorganisms from the processing environment. However, biofilms formed by pathogenic microorganisms and decaying microorganisms are unfavourable microbial reservoirs. Such microbial cells are likely to contaminate raw materials and food during processing, leading to food spoilage and economic losses to producers [4]. Pathogenic and putrefying bacteria are also major obstacles in the food industry and healthcare sector, as their ability to form biofilms shields them from ordinary cleaning procedures and allows them to persist in the environment. This persistence results in an increased microbial burden on the food processing environment and in the final food product, which further leads to spoilage and reduced shelf life, as well as increased risks from outbreaks of infectious diseases originating from food sources.

Biofilms are a substantial problem in many food processing sectors, including dairy processing, seafood processing, meat processing, food brewing, and fresh produce [5]. In the dairy processing industry, the most common bacteria associated with food contact surfaces include *Lactobacillus, Listeria*, *Enterobacter*, *Micrococcus, Bacillus*, *Staphylococcus*, *Streptococcus,* and *Pseudomonas* [6]. In the meat processing industry, the main pathogens that need to be controlled include *Staphylococcus aureus*, *Campylobacter*, *Escherichia coli* O157:H7, *Salmonella,* and *Listeria monocytogenes* [6]. In the fish product processing industry, the main pathogenic bacteria that need to be controlled include *Escherichia coli*, *Vibrio*, *Listeria monocytogenes*, *Clostridium*, *Salmonella enteritidis,* and *Staphylococcus aureus* [7]. In fact, these foodborne pathogens often form biofilms on living and abiotic surfaces during infections, resulting in cross-contamination and food safety issues.

To gain a better understanding of microbial biofilms and foodborne pathogens and determine elimination strategies to avoid food contamination, researchers have carried out a large number of studies. This review describes bacterial biofilm formation, elaborates on the problem associated with biofilms in the food industry, enumerates several kinds of common foodborne pathogens in biofilms, summarizes the current strategies used to eliminate or control harmful bacterial biofilms, introduces current and emerging control strategies, and emphasizes the future development prospects with respect to bacterial biofilms.

## 2. Overview of Biofilms

Biofilms are bacterial aggregation membranes formed by microorganisms, and these adhere to the surfaces of living or nonliving solids. Bacteria wrap themselves in an extracellular matrix by secreting extracellular polymers. In short, these biological membranes are attached to the surfaces of complex microbial communities; the microbes produce a polymer matrix consisting of extracellular polymers that are mainly composed of proteins, lipids, polysaccharides, and nucleic acids (RNA and DNA outside the cell (eDNA)), and this matrix forms a highly hydrated mixture that helps support the biofilms and their three-dimensional structures [8,9]. Biofilms can consist of individual microbial species or different combinations of protozoan, bacterial, archaeal, filamentous fungal, yeast, and algal species that form a complex three-dimensional microbiome or form extracellular polymer structures such as flocs or granules [10,11,12]. Extracellular polymers provide protection for biofilm residents by concentrating nutrients, preventing the entry of biocides, isolating metals and toxins, and preventing desiccation.

The ability of microorganisms to form biofilms has been proven to be an adaptive property of microorganisms [13]. Currently, scientists generally accept that biofilms are the primary way by which bacteria survive and grow in natural environments. Scientists at the U.S. National Institutes of Health have shown that approximately 80% of persistent bacterial infections are connected to biofilms [14]. The formation of biofilms affords a new survival mechanism, offering bacteria more suitable habitats than those of planktonic microbes. It facilitates stronger growth capacity, easier access to nutritional resources, higher survival rates when exposed to biocides, stronger capacity to evade a body’s immune system, higher biological productivity and interactions, and higher environmental stability in nutrient-poor environments [15,16,17]. Thus, biofilms provide protection for bacteria and shields them from adverse environmental pressures and antimicrobial agents under certain conditions to achieve a more favourable external environment. However, if foodborne pathogens form biofilms, the risks from foodborne disease infections can be strongly exacerbated, which can cause major public health risks and lead to adverse economic consequences. Therefore, the study of biofilms and their elimination strategies in the food industry is one of the most important research areas today.

## 3. Biofilm Formation

The formation and maturation of biofilms is a continuous, dynamic, and complex process that depends on the matrix, culture medium, intrinsic characteristics of cells, signalling molecules, cell metabolism, and genetic control [18,19]. Biofilm formation consists of five successive steps: (1) reversible attachment; (2) irreversible adhesion; (3) early development of biofilm structure (formation of small colonies); (4) biofilm maturation; and (5) cell separation and diffusion (Figure 1). The formation of bacterial biofilms begins with the uptake of organic molecules (such as proteins, lipids, polysaccharides, fatty acids, etc.) or inorganic molecules (such as inorganic salt, water, etc.) to form an appropriate surface layer, which is then embedded in a heterogeneous structure of extracellular polymers (EPS) in single or mixed communities [20]. Once bacteria are attached to a living or nonliving surface, they communicate with each other through an extracellular signalling system based on quorum sensing (QS) [21]. QS can regulate the whole stage of biofilm formation, activating certain genes in bacteria to secrete extracellular matrices, such as EPS and proteins, and gradually form a complete and mature biofilm structure. QS intercellular communication is also a controlling factor in biofilm maturation. QS is a process by which chemical communication between bacterial cells mediates the production, release, and accumulation of extracellular signal molecules. Chemical signal molecules are called autoinducers. These autoinducers are continuously produced by bacterial cells, so the level of autoinducers increases as the cell number increases (Figure 2). When autoinducers reach a minimum threshold range, these autoinducers are able to induce triggered signal transduction cascades that lead to multicellular responses in microbial populations. In other multicellular reactions, this mechanism can be involved in regulating biofilm formation, especially during the production of extracellular polysaccharides and the formation of channels or columnar structures. The formation of these structures ensures the transport of nutrients to cells in a biofilm community [22]. In addition, bacteria usually integrate the information encoded in some QS automatic induction factors into the control of gene expression to achieve mutual communication between microorganisms [23]. The normal operation of the QS system requires the participation of signal molecules, and different types of bacteria secrete different signal molecules, such as acylated homoserine lactones (AHLs) secreted by Gram-negative bacteria, autoinducing peptides (AIPs) secreted by Gram-positive bacteria, and autoinducer-2 (AI-2) secreted by both Gram-negative and Gram-positive bacteria. This shows that the regulatory mechanisms of Gram-positive and Gram-negative bacteria involved in biofilm formation may be different.

Biofilm formation is a process by which microorganisms alter their phenotype to adapt to environmental stresses or immune responses. Throughout the formation of multispecies biofilms, biofilm regulatory genes are activated and function accordingly. Furthermore, interactions between multiple species increase the possibility for biofilms to regulate genetic changes. Therefore, the formation mechanism of multispecies biofilms is intimately connected to QS, EPS, biofilm regulation genes, and additional elements. Evidence now exists that interactions between different species can significantly boost the resistance of multispecies biofilms to biocides [24]. Multispecies biofilms are characterized by greater resistance to disinfectants than single-species biofilms. Therefore, it is crucial to fully understand the mechanisms and environmental conditions that control biofilm formation to reduce the microbial risks associated with biofilm formation.

## 4. Biofilm and Food Safety

Foodborne pathogens and their biofilms are the main causes of foodborne diseases, which strongly threaten food industry development and human health. On Earth, approximately 40% to 80% of microorganisms are capable of forming biofilms [25]. Biofilms can form rapidly in food industry environments, and different microorganisms can grow on food substrates and food industry infrastructure and may lead to the formation of biofilms. In the food processing industry, microbial biofilms can appear on surfaces that come into contact with food or on surfaces that do not. In addition, a particularly important point in the food processing industry is that some biofilm-forming microorganisms that are present in food plant environments are human pathogens that can form biofilm structures on different artificial substrates, such as stainless steel, polyethylene, wood, glass, polypropylene, and rubber [26,27]. At the same time, many studies have emphasized that human immunodeficiency virus and foodborne diseases are largely caused by biofilms that form on the surfaces of equipment in the food processing and medical fields [28], and these biofilms serve as potential hosts for pathogens and are a constant source of infection and cross-contamination. Biofilms are the cause of approximately 60% of the world’s foodborne outbreaks, so the formation and presence of microbial biofilms in food processing environments is a major concern and poses risks to food safety [29,30]. In food processing environments, pollutants mainly originate from the surrounding air, equipment, or food surfaces, and if cleaning is not adequate, they are very likely to lead to the formation and accumulation of microbial biofilms, which can lead to the spoilage of food, resulting in severe public health risks and adverse economic consequences for consumers [20,29]. At the same time, biofilms also create substantial technological challenges in the food industry because biofilms may prevent heat from flowing through the surfaces of equipment, increase the frictional resistance of fluids on surfaces, lead to mechanical clogging of fluid handling systems, and increase the rate of metal surface corrosion, resulting in losses in productivity [31]. In summary, biofilms generate risks for direct pathogen contamination in the food industry, as well as the contamination of instruments and processing equipment.

Biofilms are a major challenge in the food industry, as they allow bacteria to bind to a range of surfaces, such as wood, polypropylene, glass, plastic, rubber, stainless steel, and even food, in just a few minutes; mature biofilms can then form in a matter of days (or even hours) [32]. Biofilm formation is harmful in most cases [20]. In the food industry, foodborne pathogens can form biofilms, which can lead to food spoilage, harming the health of consumers [29]. In hospital settings, biofilms can persist on surfaces of medical devices and patient tissues, resulting in persistent infection [33]. In the dairy industry, some thermophilic and cryophilic bacteria are often present during the processing, pasteurisation, and preservation stages of dairy products. For example, *Geobacillus* spp., which can grow at temperatures up to 65 °C, and its heat-resistant spores have been shown to have significant adverse effects on the production of milk powder [34]. The ability of cryophilic bacteria to thrive at refrigerated temperatures complicates the storage of dairy products, resulting in shorter shelf lives for dairy products. *Pseudomonas* was the most common psychrophilic bacterium responsible for spoilage. Without heat treatment, it can achieve high populations and form biofilms under the low-temperature conditions of milk cooling tanks and pipe walls; it can usually secrete enzymes and reduce the thermal stability of fats via protease secretion, which is a major cause of milk spoilage [35]. In addition, some examples have emerged of biofilm-related diseases in food safety. For example, the ability of the lungs of patients with cystic fibrosis to form *Pseudomonas aeruginosa* biofilms is a classic instance of biofilm involvement in chronic infections. Because *Pseudomonas aeruginosa* accumulates in biofilms, this chronic infection is usually incurable and ultimately leads to death in cystic fibrosis patients [36]. *Staphylococcus aureus* can cause food poisoning by producing enterotoxins. Biofilms can cause diseases associated with human infections, such as otitis media, bacterial vaginitis, gingivitis, dental plaque, urinary tract infection, middle ear infection, catheter, and prosthetic joint infection, contact lens infection, or cystic fibrosis [37]. Some of these infections are associated with antibiotic resistance and can be fatal, such as cystic fibrosis infection, heart valve infection, and endocarditis [38]. In addition, studies have reported that infections connected with microbial biofilms can be incredibly challenging to treat and cure because the pathogenic bacteria present in biofilm communities often exhibit strong virulence and a high degree of antimicrobial tolerance or resistance, allowing them to survive even under fairly aggressive antimicrobial therapy regimens. Therefore, it is very important to find new and effective strategies to eliminate or control the formation of biofilms that harbour harmful pathogens.

Food security is an issue of global concern. As risks from foodborne pathogen infections increase, managers of food manufacturing and processing plants must carry out more thorough and frequent assessments of pathogen growth. In addition, in light of the negative effects that pathogenic bacterial biofilms have on several areas of human health, the prevention, control, and elimination of harmful biofilms have become key issues in this field. Biofilm control methods and their uses in the food industry are briefly described in Figure 3.

## 5. Biofilm Removal and Control in the Food Industry

Globally, emphasis has been placed on biofilm formation by bacterial pathogens, particularly in the medical and food industries, because of the potential health risks and public health problems associated with biofilms. For example, biofilms not only have antimicrobial resistance and mechanical persistence but also produce virulence factors, among other substances [39,40,41], all of which can lead to severe human health problems. The growth of biofilms, which may contain food-spoiling bacteria and foodborne pathogens, in food processing environments results in an increased potential for microbial contamination of processed products. Biofilms protect the microorganisms from disinfectants, increasing their survival rate and the likelihood of the subsequent contamination of food, leading to shorter shelf lives and risks associated with disease transmission. As a result, it is necessary and important to understand and control biofilm formation and to find strategies for biofilm elimination to reduce the microbial risks associated with biofilm formation. Overall, current strategies for controlling harmful biofilms fall into three main categories [42]: (i) modifying abiotic surface features to prevent biofilm formation; (ii) regulating signalling pathways to inhibit biofilm formation and stimulate biofilm diffusion; and (iii) applying external forces to eradicate biofilms.

In the food industry, the best strategy to eradicate bacterial biofilms is to prevent biofilm formation. This can be conducted by avoiding the formation of bacterial biofilms in key locations via methods such as aseptic processing, regular disinfection cleaning, and the sterilisation of equipment after use. However, in most cases, especially during food processing, it is neither possible nor cost-effective to sterilize all environments in production areas. Therefore, other measures must be taken to decrease the population of harmful bacteria and biofilms in manufacturing areas. It is necessary to prevent the formation of biofilms by carrying out regular cleaning and disinfection so that cells do not attach firmly (reversibly) to contact surfaces. The disinfectants most widely used in sanitary disinfection programmes in the food industry are quaternary ammonium compounds (QACs), hypochlorites, peroxides (peracetic acid and hydrogen peroxide), chloramines, iodine, ozone, aldehydes (formaldehyde, glutaraldehyde, paraformaldehyde), and phenols. Today, alkyl amines, chlorine dioxide, and quaternary ammonium mixtures are also included in disinfection programmes [5]. They react with various components of bacterial cells and thus exhibit harmful effects on bacterial cells. Representative compounds for the most common disinfectants used in sanitary disinfection programmes in the food industry are briefly described in Table 1.

In the food industry, heat treatment is also a measure employed to decrease the number of harmful bacteria and biofilm populations in production areas. Steam is a promising heat treatment technology for biofilm inactivation [46,47]. Compared with other conventional heat treatments, steam heat treatment technology has the following advantages [48]. First, steam heat treatment technology can operate in an oxygen-free environment, and steam has a high heat transfer capacity during condensation. In addition, steam can easily access surface fissures or fractures in cells, thus effectively eliminating foodborne pathogens. Some studies have shown that steam pasteurisation is an efficient approach for the rapid inactivation of foodborne pathogens due to the high heat capacity [49]. Kim et al. [50] demonstrated that the inactivation of biofilms on diverse nonbiological surfaces can be accomplished extremely well using steam heat treatment technologies. The capacity of superheated steam heat treatment to destroy foodborne pathogens is large [51].

In recent years, more effective and environmentally friendly control strategies have also been discovered to eliminate or control the formation of harmful biofilms. For example, subinhibitory concentrations of ibuprofen have been demonstrated to reduce biofilm formation by *E. coli*, *Staphylococcus aureus*, *Streptococcus pneumoniae*, and *Candida albicans* on abiotic surfaces [52]. Bacteriophages and phage lysosomes have also been shown to be useful as antibiofilm agents to achieve better control of biofilm formation [42]. In contrast to phage lysosomes, phages can not only directly kill bacteria but also induce host bacterial expression of EPS degradation enzymes, thus accelerating the removal of mature biofilms [53]. Furthermore, combined techniques can also be used in which multiple bacteriophages or phage lysates are employed to achieve a broad spectrum of antibacterial effects. In addition, the development or research into other physical surface decontamination technologies for the eradication of bacteria from biofilms has become increasingly popular in recent years, including photodynamic inactivation using pulsed ultraviolet light, electron beam irradiation, steam heating, irradiation at 405 nm, or surface treatment with ozone, ultrasound, or gaseous chlorine dioxide [54]. Several novel biofilm eliminations and control methods used in the food industry are briefly summarized in Table 2. All of these cutting-edge methods hold out hope for the future in preventing biofilm formation in the food industry.

These techniques are now the subject of ongoing studies, but we think that such products will undoubtedly become available soon. We also hope that researchers can uncover the complete mechanisms of biofilm antibiotic resistance in the near future to lay a solid foundation for the design and development of new biofilm antibiotics.

## 6. Summary of Common Foodborne Pathogens and Their Implication for Food Safety

Food contamination by pathogenic microorganisms has developed into a significant public health issue and has resulted in significant economic losses worldwide. Foodborne pathogenic bacteria can adhere to food by forming biofilms and survive for long periods of time on surfaces that come in contact with food, resulting in postprocessing contamination, a reduction in product quality and shelf life, and potential disease transmission, which are significant food safety problems in the food industry.

A summary of common foodborne pathogens and their implications for food safety are briefly described in Table 3. In the following sections, we discuss the food safety and clinical aspects connected with the five most significant foodborne bacterial pathogens (*Listeria monocytogenes*, *Salmonella enteritidis*, *Pseudomonas aeruginosa*, *Staphylococcus aureus*, and *Escherichia coli*), as well as their ability to form biofilms on various surfaces.

### 6.1. Listeria monocytogenes

Foodborne pathogens can adhere to food items by forming biofilms and survive for long periods of time on surfaces that come in contact with food, and this introduces significant issues related to food safety in the food industry. *Listeria monocytogenes* is the main pathogen related to foodborne diseases around the world and has a high fatality rate and hospitalisation rate [77]. *Listeria monocytogenes* is a gram-positive, aerobic, nonsporoforming, rod-shaped bacterium that belongs to the genus Listeria firmicide. Out of the 17 listeria species described, it is the only pathogenic species and is the pathogenic factor in listeriosis [78]. It can infect a large number of host organs, such as the liver, spleen, cerebrospinal fluid, and blood, among which the liver is the main site of infection [79]. Meningitis, sepsis, and other central nervous system infections are common in Listeria patients. In healthy adults, diarrhea and fever are the primary symptoms [80]. In pregnant women, listeriosis may cause fever, diarrhoea, spontaneous abortion, or stillbirth [81]. In neonates, listeriosis can cause septicemia, pneumonia, and meningitis [82,83]. *Listeria monocytogenes* may potentially cause a noninvasive illness commonly referred to as febrile gastroenteritis or noninvasive gastroenteritis, which has been associated with contaminated deli meats, chocolate milk, cheese, smoked fish and corn [84]. Pregnant women, foetuses or newborns, the elderly, and people with compromised immune systems are at high risk for developing diseases such as sepsis, meningitis, or gastroenteritis. In general, aminopenicillin or benzylpenicillin alone or in combination with aminoglycosides are the antibiotics typically recommended for *Listeria monocytogenes* infection [77,85,86].

The pathogenic factors in listeriosis are ubiquitous in nature and can invade the food processing environment. According to a two-year survey conducted by Wu et al. [87], *Listeria monocytogenes* had the highest contamination rate in China’s food industry, accounting for approximately 20%. Additionally, 99% of listeriosis cases were brought on by the consumption of food tainted with *Listeria monocytogenes*, and only a small number of cases were brought on by pathogens found in the environment [88]. At the same time, food processing contamination is the primary transmission route for *Listeria monocytogenes* [77]. Therefore, the first line of defence in the control and prevention of listeriosis is routine cleaning, disinfection processes, and the application of appropriate food hygiene standards during food preparation. Quaternary ammonium compounds and chlorine-based biocides are the two biocides most often utilized for *Listeria monocytogenes* [89], when used at recommended dosages, are highly effective against *Listeria monocytogenes*.

### 6.2. Salmonella enterica

*Salmonella enterica* is among the most prevalent foodborne pathogens worldwide and has been linked to high-profile outbreaks in many foods. It has two species, *S. Enterica* and *S. Bongori*, and more than 2500 known serum variants, and it is a gram-negative, facultatively anaerobic, flagellated enterobacter. *Salmonella* is a human and animal pathogen that causes salmonellosis, which is the most typical (85%) foodborne illness [90]. Salmonellosis is an infectious disease, and the pathogen responsible causes human illnesses that begins with gastroenteritis and end with systemic infections. Approximately 99.5% of all isolates of pathogenic *Salmonella* in humans and other warm-blooded animals were of the species *Salmonella enterica* [91,92]. Therefore, *Salmonella enterica* is among the most important intestinal bacterial foodborne pathogens [93]. Salmonellosis usually manifests as gastroenteritis, accompanied by fever, diarrhoea, and abdominal cramps. The symptoms of salmonellosis are usually mild and can be cured without therapy within 1–4 days, but in severe cases, salmonellosis can lead to acute gastroenteritis, food poisoning, sepsis, etc. [94]. The severity of disease manifestation depends on a patient’s susceptibility to the pathogen and the virulence of the particular serum variant. Due to the widespread incidence and severity of this disease, the prevalence of *Salmonella enterica* in a country’s food supply has been considered an important benchmark for public health [95].

The major source of *Salmonella enterica* infections in humans is food, such as eggs, egg products, and poultry meat [75,96]. Contaminated food is the primary mode of transmission of *Salmonella enteritidis*, which has a high survival rate and can thrive on undercooked or improperly stored meat and animal products. In some instances, *Salmonella* infections can sometimes persist for several years without showing any overt clinical symptoms in both people or animals. Therefore, salmonellosis is a significant issue for human health because large numbers of animal hosts exist, transmission is easy, and carriers can be asymptomatic. In addition, the persistence of Salmonella in the food industry is a primary food health issue, as it can form biofilms in food processing environments and become a potential host for food contamination. Therefore, *Salmonella* produces biofilms, which are key components of its pathogenicity. *Salmonella*, similar to other bacterial pathogens, can exist in a wide range of cell surface structures (particularly those with protein-like and carbohydrate-like properties), which may enable effective aggregation of the bacterial cells with those of other species and thus promote the formation of single or multiple biofilm cell communities. *Salmonella* biofilms can exist not only on biological surfaces but also on abiotic surfaces such as concrete, stainless steel, ceramic tile, glass, granite, quartz stone, rubber, and synthetic plastics [97,98]. By encouraging the creation of virulence factors, and due to the mechanical resistance and antimicrobial resistance components of biofilms, they improve the odds of microbial survival [39].

### 6.3. Staphylococcus aureus

*Staphylococcus aureus* is one of the most common foodborne pathogens related to food safety problems [99]. *Staphylococcus aureus* is an enterotoxin-producing gram-positive bacterium that is often parasitic on the skin, throat, intestines, stomach, nasal cavity, carbuncle, and sores of humans and animals. It is a zoonotic pathogen that can lead to cardiovascular infection, surgical site infection, lower respiratory tract site infection, cystic fibrosis pulmonary infection, endocarditis, and pneumonia in humans and animals [100,101]. Additionally, *Staphylococcus aureus* is a highly adaptive microbe that can live in a variety of environments (such as air, sewage, and soil) by forming biofilms. Moreover, *Staphylococcus aureus* is highly capable of forming biofilms on the surfaces of food, on food processing equipment, and in water, which are sources of cross-contamination of food [102]. The control of *Staphylococcus aureus* in environments where food is processed is also complicated by its propensity to adhere to food-contacting surfaces and form biofilms. The ability of this species to form biofilms and achieve cell adhesion is connected to the production of polysaccharide intercellular adhesion (PIA). PIA is encoded by the ICA operon, which contains the icaADBC gene cluster. These four genes encode proteins that mediate PIA synthesis and elongation [103].

The World Health Organization has classified *Staphylococcus aureus* as a high-priority species on its list of antibiotic-resistant bacteria that are dangerous to human health [104]. *Staphylococcus aureus* forms biofilms as one of its most effective survival strategies; thus, it is difficult to treat even with antibiotics and causes a severe burden in medical settings. The primary issue related to *Staphylococcus aureus* biofilms is their resistance to antibiotics and their host defence mechanisms, and the biofilms’ properties confer increased resistance to *Staphylococcus aureus* pathogenic strains to antibiotics and host defence factors [105]. This resistance is mainly achieved through the diffusion barrier formed by the polysaccharide matrix [106]. Therefore, the emergence of MRSA strains is an issue in public health, and due to the formation of *Staphylococcus aureus* biofilms, bacterial sensitivity to antibiotics and even to vancomycin has decreased, making the removal of *Staphylococcus aureus* difficult [107].

*Staphylococcus aureus* is a pathogenic bacterium that colonizes 30% to 50% of healthy people, and it can adhere to surfaces such as glass, metal, plastic, and host tissues. Infections associated with *Staphylococcus aureus* biofilms include osteomyelitis, endocarditis, chronic wound infections, eye infections, multimicrobial biofilm infections, and kidney abscesses [108]. In general, to prevent *Staphylococcus aureus* from adhering to biological or abiotic surfaces, anti-adhesive agents such as calcium chelators, silver nanoparticles, aryl rhodamine, and chitosan can be applied to surfaces [107]. In addition, nucleases, proteases, dispersin B, lysococcin, and hyaluronic acid lysase can disrupt and inhibit biofilms through different biofilm dispersal mechanisms.

### 6.4. Pseudomonas aeruginosa

One of the most virulent pathogens, *Pseudomonas aeruginosa*, is a major contributor to a number of acute infections [66]. In 2017, the World Health Organization designated it as a pathogen that requires high-priority research and the development of new medications [67]. *Pseudomonas aeruginosa*, which belongs to the genus Pseudomonas, is an aerobic gram-negative opportunistic pathogen. It is also an important water source and conditional foodborne pathogenic bacterium that is widely distributed and resistant to adverse environments. It mainly exists in soil, dust, and water and small numbers in human intestines. It is mainly parasitic in the genital parts, anus, external auditory canal, and armpit, and can also temporarily parasitize skin surfaces. *Pseudomonas aeruginosa* has many pathogenic factors and is a completely pathogenic bacterium that leads to human acute intestinal diseases and skin inflammation. In addition, in individuals with severe conditions as well as those who have burns, surgical wounds, foot ulcers, and diabetes, *Pseudomonas aeruginosa* can lead to severe acute and chronic infections; for example, *Pseudomonas aeruginosa* is a significant contributor to cystic fibrosis [109]. *Pseudomonas aeruginosa* infection can also occur in healthy individuals, causing external auditory canal inflammation, otitis media, keratitis, and folliculitis [110]. If not properly treated in the acute state, *Pseudomonas aeruginosa* can form biofilms, establishing a chronic biofilm infection that is difficult to eradicate. Biofilm formation is certainly one of the most significant factors affecting virulence in the pathogenesis of *Pseudomonas aeruginosa* infections [111]. Biofilms allow these pathogens to attach to different surfaces, providing protection from severe environmental factors and the immunological systems of hosts. In addition to these basic protections, biofilms provide microbes with a safe haven for antibiotic resistance in vivo, leading to the emergence of the MDR phenotype. Therefore, biofilm formation is a key reason that *Pseudomonas aeruginosa* has become a hospital pathogen and is an important indicator of the persistence of chronic bacteria. In addition, *Pseudomonas aeruginosa* has a distinct advantage: it can move and easily travel from one niche to another without difficulty [111]. Three movement types have been observed, namely, smattering motion, swimming motion and convulsive motion, which enable *Pseudomonas aeruginosa* to exist in various environments [112].

Currently, controlling *Pseudomonas aeruginosa* infections is enormously challenging because of the emergence of antibiotic-resistant strains. *Pseudomonas aeruginosa* is the most frequent pathogen detected in hospital-associated infections (HAI) and is the second-most frequent cause of ventilator-associated pneumonia in the United States [110]. On the one hand, *Pseudomonas aeruginosa* is capable of producing a variety of virulence factors, including elastase, flagella, alkaline protease, type IV pili, lipopolysaccharide, exotoxin A, phospholipase, pyocyanin, pyochelin, pyoverdine, and *Pseudomonas* quinolone signal (PQS) [113]. On the other hand, *Pseudomonas aeruginosa*, has a genome that is relatively large compared with that of other prokaryotes and has abnormal chromosome regulation of genes, which helps the species adapt to various environmental conditions, thus increasing the incidence of disease and mortality, and it is intimately connected to the rise of antibiotic resistance [114,115]. Another cause of *Pseudomonas aeruginosa* resistance is the formation of biofilms. Because biofilms are not degraded by antimicrobial agents (such as disinfectants), heat, or drying and remain on living or abiotic surfaces, especially in hospitals, they can lead to contamination and the spread of infectious illnesses. Therefore, biofilms are a key contributor to infectious illnesses because they act as a barrier between the immune system and antibiotic drugs [116]. *Pseudomonas aeruginosa* plays a major role in hospital-acquired infections, particularly in burn patients, so knowledge of these strains is of particular epidemiological importance for the prevention and control of *Pseudomonas aeruginosa* infections.

### 6.5. Escherichia coli

*Escherichia coli* is common in both humans and animals as part of the regular flora and is generally harmless to humans [117]. However, the development of virulence factors causes some strains of *E. coli* to become pathogenic, and as a result, they rank among one of the most prevalent foodborne pathogens associated with food safety issues. *E. coli* is a gram-negative, nonsporoforming, metabolically active, rod-shaped bacterium. A few special serotypes of *E. coli* exhibit pathogenicity, and according to their different pathogenicity, they can be roughly divided into enteroaggregative *E. coli* (EAEC), enterohemorrhagic *E. coli* (EHEC), enteroinvasive *E. coli* (EIEC), enteropathogenic *E. coli* (EPEC), enterotoxigenic *E. coli* (ETEC), and diffuse-adhering *E. coli* (DAEC), which commonly manifest in human infections as diarrhoeal illness. Among them, EHEC can produce Shiga toxin, which can cause diarrhoea, haemorrhagic enteritis, haemolytic uraemic syndrome (HUS), thrombotic thrombocytopenic purpura (TTP), and other diseases through foodborne infections [118]. Children, the elderly, and immunocompromised people may even develop systemic infections or acute renal failure. Meningitis, sepsis, and urinary tract infections are frequently ascribed to extraintestinal *E. coli* pathotype infections, including neonatal meningitis-associated *E. coli* (NMEC), sepsis-associated *E. coli* (SEPEC), and uropathogenic *E. coli* (UPEC), respectively [117].

Because the presence of *E. coli* implies unsanitary conditions in the food industry, it serves as a hygiene indicator. Regulation 2073/2005 of the European Commission states that the amount of *E. coli* that can be found in certain meat products (such as minced beef) cannot be more than 500 CFU/g [119]. The main pathogenic strain of *E. coli* is EHEC- O157:H7, which can cause infectious diarrhoea and haemorrhagic enteritis. It mainly causes human infection through contaminated food, including fresh meat, fruits, vegetables, raw milk, and dairy products. *E. coli* O157:H7 has a strong pathogenic capacity and is resistance to gastric acid, and it is destructive to cells. *E. coli*, similar to most foodborne microbes, can form biofilms by adhering to a range of food-contact surfaces. Biofilms are more resistant to environmental stresses, such as UV light exposure, sanitising agents, nutritional and oxidative stresses, and desiccation. Consequently, biofilms are important for both public health and the economy because they cause chronic illnesses that are challenging to cure and are resistant to cleaning and sanitation [120].

## 7. Conclusions and Future Perspectives

The majority of bacteria in nature are found in biofilms. Biofilm properties provide protection against environmental pressures and enhance resistance to antimicrobial agents, contributing to microbial persistence and toxicity. Antimicrobial resistance is significantly increased when bacteria form biofilms, so the formation of bacterial biofilms is a significant cause of many persistent and chronic infectious diseases. In addition, infections associated with bacterial biofilms are challenging to treat and are highly resistant to both host immune systems and antibiotics, which poses a substantial challenge for the treatment of biofilm-associated infections.

Food poisoning is a general term for illnesses caused by the consumption of foods typically contaminated with bacteria, viruses, toxins, or parasites. Food is rich in nutrition and is thus suitable for the growth and reproduction of pathogens; therefore, microbial biofilms easily form on food. Microbial biofilms may contain a significant number of decaying and pathogenic microorganisms, so the presence of biofilms on surfaces that come into contact with food is generally considered harmful to human health. The economic losses incurred during epidemics of foodborne pathogens mean that the formation and presence of bacterial biofilms can have a significant influence on businesses that process food, so impeding their capacity to survive under these circumstances is a particularly appealing goal for both food sector workers and researchers.

Although research on bacterial biofilms is advancing at present, the bacterial biofilm formation mechanisms still need further study. Understanding the specifics of biofilm formation and how signalling pathways are regulated by biofilm formation can help us identify novel targets for the development of highly effective small peptide or protein inhibitors that have strong antibiofilm properties. Additionally, as we continue to explore the mechanisms underlying the biofilm life cycle in the future, more powerful antibiofilm drugs may be discovered, and synthetic derivatives with structural modifications may be designed to create more powerful inhibitors or alter the way we apply them to achieve more effective and rapid suppression or elimination of harmful biological membranes.

Today, traditional control strategies, including mechanical and manual cleaning, chemical cleaning, aseptic processing, periodic disinfection cleaning, final sterilisation of equipment, and heat treatment, are still in use and are being further developed. To satisfy the requirements for food safety set out by the food processing industry, more effective and ecologically friendly control strategies must be developed owing to the rising resistance of biofilms to traditional disinfection procedures. In recent years, it has become increasingly popular to develop or investigate emerging strategies for controlling or eliminating biofilms, which include the use of enzyme treatments, phage treatments, pulsed ultraviolet light treatments, steam heat treatment technologies, cold plasma technologies, electron beam irradiation technologies, irradiation at 405 nm, or surface treatment with ozone, ultrasound, or gaseous chlorine dioxide. Different treatments for biofilms at different periods will make the removal of biofilms more efficient, so these emerging biofilm control strategies can provide a new, diverse, and targeted solution for food safety. In addition, biofilm inhibition and QS by natural biological agents will also help to address biofilm issues.

Studies have already demonstrated that the extracellular polymers produced by bacteria can sustain the high osmotic pressure inside biofilms, improving the biofilm’s capacity to absorb nutrients from the environment and fuel biofilm proliferation. Therefore, future studies could focus on methods to regulate the osmotic pressure of biofilms to eliminate or control the formation of hazardous biofilms while simultaneously promoting the formation of beneficial biofilms. In addition, combined technologies, which integrate two or more different control technologies, are promising new approaches to eliminate or control the formation of harmful biofilms. For instance, a combination of chemical agents and UV irradiation can effectively remove *Pseudomonas aeruginosa* biofilms. Combined technologies not only provide synergistic effects but also reduce material and energy consumption. Therefore, the development of combined technologies to eliminate or control the formation of harmful biofilms in the future may also become an attractive research focus.

Future research into biofilm control will require a multidisciplinary approach, and although there may be many difficulties, these will be overcome as the research progresses. We hope that this summary will serve as a reference and provide effective strategies for the prevention, suppression, and even eradication of biofilms.

## Figures and Tables

**Figure 1 molecules-28-02432-f001:**
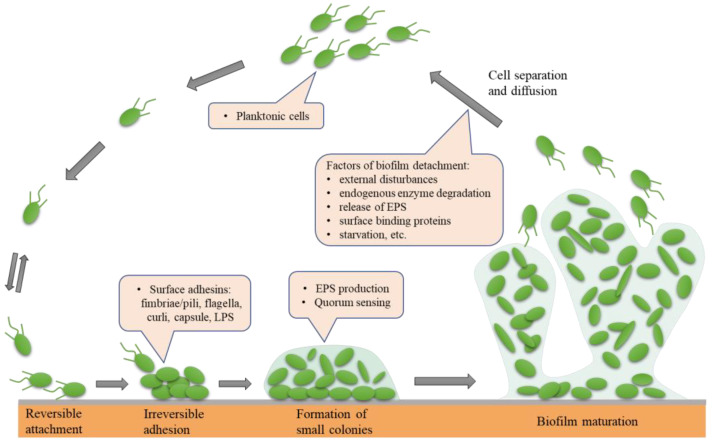
The five main phases leading to the development and formation of biofilm: (1) reversible attachment; (2) irreversible adhesion; (3) early development of biofilm structure (formation of small colonies); (4) biofilm maturation; (5) cell separation and diffusion.

**Figure 2 molecules-28-02432-f002:**
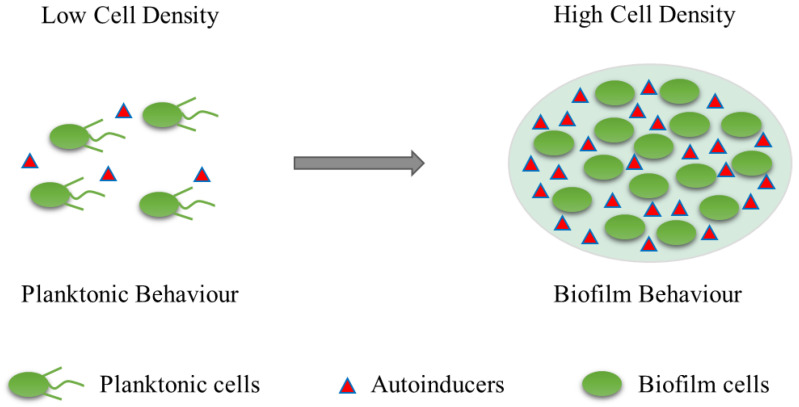
Quorum sensing (QS) illustration. Autoinducer is a signaling molecule. These autoinducers are continuously produced by bacterial cells, so the level of autoinducers increases as the cell number increases.

**Figure 3 molecules-28-02432-f003:**
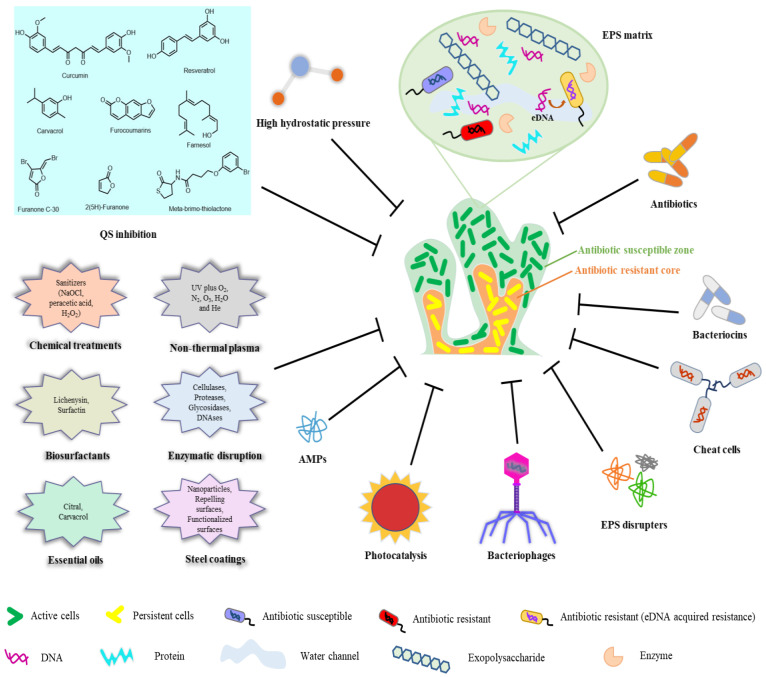
Biofilm control methods and their uses in the food industry.

**Table 1 molecules-28-02432-t001:** Representative compounds for the most common disinfectants used in sanitary disinfection programmes in the food industry.

Disinfectants	Characteristics	Function	Types of Microorganisms Acting	References
Sodium hypochlorite (NaClO)	strong oxidising agents	Eliminate cells given their ability to cross the cell membrane, and to oxidise the sulfhydryl groups of certain enzymes participating in the glycolytic pathway. Can react with wide-ranging biological molecules under physiological pH conditions, such as proteins, amino acids, lipids, peptides, and DNA.	*Staphylococcus aureus*, *Prevotella intermedia*, *Peptostreptococcus miros*, *Streptococcus intermedius*, *Fusobacterium nucleatum*, *Enterococcus faecalis*, *Listeria monocytogenes*, *Pseudomonas fragi*, *Staphylococcus xylosus*, *Bacillus cereus*	[20,43]
Quaternary ammonium (QACs)	surface-active agents, membrane-active agents, hydrophobic activity	Reduces surface tension and forms micelles, leading to dispersion in the liquid to facilitate microbial removal. Interact with not only the cytoplasmic membrane of bacteria but with the plasma membrane of yeast. effective against lipid-containing viruses. Interact with intracellular targets and bind to DNA.	*Listeria monocytogenes*, *Bacillus cereus*, *Staphylococcus* spp., *Pseudomonas* spp.	[20,32]
Peracetic acid (PAA)	strong oxidising agents	Capable of causing the oxidation of thiol groups in proteins, disruption of membranes, or damage to bases in DNA.	*Listeria monocytogenes*, *Staphylococcus aureus*, *Pseudomonas aeruginosa*	[32,44]
Hydrogen peroxide (H_2_O_2_)	highly oxidising capacity	Production of free radicals affecting the biofilm matrix.	*Staphylococcus aureus*, *Pseudomonas aeruginosa*, *Vibrio* spp.,	[5,43,45]

**Table 2 molecules-28-02432-t002:** Several novel biofilm eliminations and control methods are used in the food industry.

Methodology	Mechanism of Action	Description	Reference
Electrolyzed water	Promote biofilm dispersion	acidic and slightly acidified electrolyzed water can efficiently remove *L. innocua*, *L. monocytogenes*, *Vibrio parahaemolyticus*, *E. coli*, and *B. cereus* biofilms.	[54]
Bacteriophages	Cell lysis	can not only directly kill bacteria, but also induce host bacteria to express EPS degradation enzymes, thus accelerating the clearance of mature biofilms.	[53]
Nonthermal atmospheric plasmas	Bactericidal	demonstrated high disinfectant capacity, contact-free and waterless, over conventional chemical-based disinfection.	[54]
Bacteriocins	Cell membrane alteration	Such as the bacteriocins nisin, subtilomycin, lichenicidin, enterocin B3A-B3B, enterocin AS-48, and sonorensin.	[54]
Biosurfactants	Inhibition of bacterial adhesion	Avoid biofilm formation and even inhibit QS molecules	[55]
Enzymatic disruption	Extracellular matrix disruption	Such as cellulases, proteases, glycosidases, and DNAses.	[29]
QS inhibition	Downregulation of adhesion and virulence mechanisms	Binding of inhibitors to QS receptors (lactic acid), enzymatic degradation of QS signals (paroxonases), sRNA post-transcriptional control, inhibition of QS signals biosynthesis.	[29]
High hydrostatic pressure	Bactericidal and endospores removal	high hydrostatic pressure (up to 900 MPa) combined with thermal treatments (50–100 °C)	[29]
Novel physical microbial inactivation technologies	Inactivation of microorganisms within biofilms	Such as photodynamic inactivation using pulsed ultraviolet light, electron beam irradiation, steam heating, light at 405 nm, and treatment of the surfaces using ozone, ultrasounds, and gaseous chlorine dioxide.	[54]

**Table 3 molecules-28-02432-t003:** A summary of common foodborne pathogens and their implications for food safety.

Foodborne Pathogens	Characteristics	Contaminated Food	The Main Symptoms of FOOD Poisoning	Examples of Harmful Spoilage Effects	References
*Listeria monocytogenes*	Gram-positive, rod-shaped, facultative anaerobic, non-spore forming	meat (especially beef), eggs, poultry, seafood, vegetable, salad, juice, milk, cheese, dairy, ice-cream	diarrhea and fever	meningitis, encephalitis, endocarditis, sepsis, pneumonia, and other central nervous system infections	[56,57,58]
*Salmonella enterica*	Gram-negative, rod-shaped, facultative anaerobic, flagellate, non-spore forming	eggs, egg products, poultry meat	fever, diarrhea, and abdominal cramps	gastroenteritis, sepsis	[59,60,61,62]
*Staphylococcus aureus*	Gram-positive, spherical, facultative anaerobic, flagellate, non-spore forming, non-motile	meat products, dairy products, egg products, poultry, salads, bakery products (especially cream-filled pastries and cakes, and sandwich fillings)	nausea, vomiting, spasmodic pain in the middle and upper abdomen, diarrhea	osteomyelitis, endocarditis, chronic wound infection, eye infection, multimicrobial biofilm infection, renal abscess	[63,64]
*Pseudomonas aeruginosa*	Gram-negative, rod-shaped, obligate aerobic, flagellate, motile	fruits, vegetables, meat, low-acid dairy products	fever, ulceration, diarrhea, expectoration	postoperative wound infection, urinary tract infection, bedsore, abscess, external otitis, otitis media, keratitis, folliculitis, sepsis, cystic fibrosis	[65,66,67]
*Escherichia coli*	Gram-negative, rod-shaped, non-spore forming, metabolically active	fresh meat, fruits, vegetables, raw milk, dairy products	nausea, vomiting, abdominal cramps, bloody diarrhea, fever	gastrointestinal infections, urinary tract infections, septic infections, hemorrhagic colitis, hemolytic uremic syndrome, thrombotic thrombocytopenic purpura, kidney failure	[68,69,70,71]
*Bacillus cereus*	Gram-positive, rod-shaped, facultative aerobic, spore-forming, motile	dairy products, vegetables, meat, rice	diarrhoea and vomiting symptoms	meningitis, brain abscess, cellulitis, endophthalmitis, pneumonia, endocarditis, and osteomyelitis	[72,73,74]
*Campylobacter jejuni*	Gram-negative, rod-shaped, microaerophilic, flagellate, non-spore forming, motile	Animals, poultry, vegetables, fruits, all kinds of cooked food, milk, dairy products	bloody diarrhoea, fever, stomach cramps, nausea, and vomiting	gastrointestinal infection, acute enteritis, septicemia, meningitis, arthritis, pyelonephritis	[75,76]

## Data Availability

Not applicable.

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
