# Peer review of "Biofilm Formation and Control of Foodborne Pathogenic Bacteria"

_molecules, 2023, doi:10.3390/molecules28062432_

Round 1

Reviewer 1 Report

The manuscript entitled " Biofilm formation and control of foodborne pathogenic bacteria" has discussed the impact of pathogenic bacteria and their biofilm process on food items and their severity in the food sector.  Also, the importance of various methods employed to overcome the pathogenicity of the microorganisms. 

  Comments: The review focuses much more generally on the bacteria and the various techniques used in the food industry to reduce the pathogenesis. Already various reviews discuss a similar fashion regarding the pathogenesis of bacteria in the food industry. And also already one review article with almost similar kind of information has been published by the same author entitled "Biofilm formation and control strategies of foodborne pathogens: food safety perspectives". What is the novelty of this review article?

 1. Discuss the Quorum sensing mechanism and its involvement in the regulation of biofilm formation briefly. 

2. What are the other possible mechanism bacteria adopt and release virulence molecules in the environment (food items)? (If possible draw some diagrammatic representation of the mechanism)

3. Differentiate the mechanism and their significance for the gram-positive and gram-negative bacteria.

 4. Explain the different methodologies such as physical, chemical, and biological to counter or reduce the pathogenesis (particularly biofilm formation) of the bacteria during their presence in the food items.

5. Also, discuss the recent trend of the techniques used for the removal or prevention of biofilms in food items.

 Grammar corrections have to be done since there are some phrase errors.

Author Response

Response to Reviewer 1 Comments

Comments: The review focuses much more generally on the bacteria and the various techniques used in the food industry to reduce the pathogenesis. Already various reviews discuss a similar fashion regarding the pathogenesis of bacteria in the food industry. And also already one review article with almost similar kind of information has been published by the same author entitled "Biofilm formation and control strategies of foodborne pathogens: food safety perspectives". What is the novelty of this review article?What is the novelty of this review article?

Response:

Thanks for your comments. Compared to the article “Biofilm formation and control strategies of foodborne pathogens: food safety perspectives”, this article focuses more on the food safety risks posed by some typical foodborne pathogens (food contamination categories, food poisoning symptoms and illness), emerging biofilm control strategies and food safety issues, and less on the aspects of biofilm formation. This paper tends to use tables for summary, with complete documents and more information, which makes it clear and easy to read. In our review, we are also good at summarizing its important contents in the form of figures, such as the development process of biofilm formation, quorum sensing illustration, and biofilm control methods.

  1. Discuss the Quorum sensing mechanism and its involvement in the regulation of biofilm formation briefly.

Response:

Thanks for your correction. We have added the relevant content to the article (Line 125-127), and here is the overall answer to this question.

Quorum sensing can regulate the whole stage of biofilm formation, activating certain genes in bacteria to secrete extracellular matrix, such as EPS and proteins, and gradually form a complete and mature biofilm structure. QS intercellular communication is also a controlling factor in biofilm maturation. QS is a process by which chemical communication between bacterial cells mediates the production, release and accumulation of extracellular signal molecules, known as autoinducers. These autoinducers are continuously produced by bacterial cells, so the level of autoinducers increases as the cell density increases. For high population densities, these autoinducers are able to induce triggered signal transduction cascades that lead to multicellular responses in microbial populations. In other multicellular reactions, this mechanism can be involved in regulating biofilm formation, especially during the pro-duction of extracellular polysaccharides and the formation of channels or columnar structures. The formation of these structures ensures the transport of nutrients to cells in a biofilm community.

  1. What are the other possible mechanism bacteria adopt and release virulence molecules in the environment (food items)? (If possible draw some diagrammatic representation of the mechanism)

Response:

Thanks for your correction. Here's our answer to the question.

Possible mechanisms by which bacteria adopt and release virulence molecules in the environment (food items) include biofilm formation, quorum sensing, exopolysaccharide modification, etc. Biofilm formation is one of the ways, which we have described as the main content in this review. However, other possible mechanisms of bacterial regulation of virulence factors have not become the focus of this paper. This article should not change the structure of the whole article and deviate from the focus. If you are interested, you can refer to references [1–5]. Next we have briefly described two other possible mechanisms (quorum sensing and exopolysaccharide modification) as below.

Quorum sensing: Quorum sensing is a cell-to-cell communication system that exists widely in the microbiome and is related to cell density. The signaling molecules accumulate in the surrounding environment with an increase of bacterial density. When the concentration of signaling molecules reaches a minimal threshold, they bind to receptor proteins, thereby activating a variety of downstream cellular processes including virulence and drug resistance mechanisms, tolerate antibiotics, and harm the host. In summary, bacteria use QS to regulate diverse arrays of functions, including virulence and biofilm formation.

Exopolysaccharide modification: Exopolysaccharide modifications have a key role in bacterial biofilm formation, immune evasion and virulence. Production of an extracellular mixture of sugar polymers called exopolysaccharide is characteristic and critical for biofilm formation. Furthermore, the biosynthesis and modification of extracellular polysaccharide components can influence bacterial pathogenicity and virulence. For example, Staphylococcus epidermidis is an important human pathogen that frequently causes persistent infections by biofilm formation on indwelling medical devices. It produces a poly-N-acetylglucosamine molecule that emerges as an exopolysaccharide component of many bacterial pathogens. Deacetylation of poly-N-acetylglucosamine molecule is essential for key virulence mechanisms in Staphylococcus epidermidis, namely biofilm formation, settlement and resistance to neutrophil phagocytosis and human antibacterial peptides.

References:

  1. Li, Y.H.; Tian, X. Quorum Sensing and Bacterial Social Interactions in Biofilms. Sensors 2012, 12, 2519–2538, doi:10.3390/s120302519.
  2. Zhao, X.; Yu, Z.; Ding, T. Quorum-Sensing Regulation of Antimicrobial Resistance in Bacteria. Microorganisms 2020, 8, 425, doi:10.3390/microorganisms8030425.
  3. Zhou, L.; Zhang, Y.; Ge, Y.; Zhu, X.; Pan, J. Regulatory Mechanisms and Promising Applications of Quorum Sensing-Inhibiting Agents in Control of Bacterial Biofilm Formation. Front. Microbiol. 2020, 11, 1–11, doi:10.3389/fmicb.2020.589640.
  4. Vuong, C.; Kocianova, S.; Voyich, J.M.; Yao, Y.; Fischer, E.R.; DeLeo, F.R.; Otto, M. A Crucial Role for Exopolysaccharide Modification in Bacterial Biofilm Formation, Immune Evasion, and Virulence. J. Biol. Chem. 2004, 279, 54881–54886, doi:10.1074/jbc.M411374200.
  5. Koo, H.; Falsetta, M.L.; Klein, M.I. The Exopolysaccharide Matrix: A Virulence Determinant of Cariogenic Biofilm. J. Dent. Res. 2013, 92, 1065–1073, doi:10.1177/0022034513504218.

  1. Differentiate the mechanism and their significance for the gram-positive and gram-negative bacteria.

Response:

Thanks for your correction. We have added the following content in the revised manuscript as below (Line 142-148).

The normal operation of the QS system requires the participation of signal molecules, and different types of bacteria secrete different signal molecules, such as acylated homoserine lactones (AHLs) secreted by Gram-negative bacteria, autoinducing peptides (AIPs) secreted by Gram-positive bacteria, and autoinducer-2 (AI-2) secreted by both Gram-negative and Gram-positive bacteria. This shows that the regulatory mechanisms of Gram-positive and Gram-negative bacteria involved in biofilm formation may be different.

  1. Explain the different methodologies such as physical, chemical, and biological to counter or reduce the pathogenesis (particularly biofilm formation) of the bacteria during their presence in the food items.

Response:

Thanks for your comments. Here's our answer to the question.

Physical method: In the food industry, food safety can be achieved by preventing the formation of bacterial biofilms in key areas via methods such as aseptic processing, regular disinfection cleaning and the sterilization of equipment after use. It is necessary to prevent the formation of biofilms by carrying out regular cleaning and sterilization so that cells do not attach firmly (reversibly) to contact surfaces. In addition, the development or research into other physical surface decontamination technologies for the eradication of bacteria from biofilms has become increasingly popular in recent years, including photodynamic inactivation using pulsed ultraviolet light, steam heat treatment, irradiation at 405 nm, or surface treatment with ozone, ultrasound, or gaseous chlorine dioxide.

Chemical method: Prevent the formation of bacterial biofilms by using disinfectants. The disinfectants most widely used in sanitary disinfection programmes in the food industry are quaternary ammonium compounds (QACs), hypochlorites, peroxides (peracetic acid and hydrogen peroxide), chloramines, iodine, ozone, aldehydes (for-maldehyde, glutaraldehyde, paraformaldehyde) and phenols. Today, alkyl amines, chlorine dioxide and quaternary ammonium mixtures are also included in disinfection programmes. They react with various components of bacterial cells and thus exhibit harmful effects on bacterial cells.

Biological method: Bacteriophages and phage lysosomes have also been shown to be useful as antibiofilm agents to achieve better control of biofilm formation. In contrast to phage lysosomes, phages can not only directly kill bacteria but also induce host bacterial expression of EPS degradation enzymes, thus accelerating the removal of mature biofilms. Furthermore, combined techniques can also be used in which multiple bacteriophages or phage lysates are employed to achieve broad spectrum of antibacterial effects.

All of the above content is reflected in the article, but we are in a sequential order to elaborate, so that more can reflect the development of the method of eliminating or reducing biofilm.

  1. Also, discuss the recent trend of the techniques used for the removal or prevention of biofilms in food items.

Response:

Thanks for your comments and correction. We have added to the "Conclusions and future perspectives" section to discuss the recent trend of the techniques used for the removal or prevention of biofilms in food items (Line 569-578).

Grammar corrections have to be done since there are some phrase errors.

Response:

The manuscript has been reviewed and edited for proper English language by a native English speaker.

Reviewer 2 Report

Dear Editor and authors!

I evaluated the review „Biofilm formation and control of foodborne pathogenic bacteria” from Xiaoli Liu et al. The authors nicely summarize the recent state of the art, but, at least a little bit, they neglecting a promised outlook. I will describe my remarks in a greater detail below.

Line 8 – 10: You are using the words microbial and extracellular a bit excessive. For instance, The ECM of biofilms is primarily composed of extracellular polysaccharides, extracellular proteins, and extracellular DNA”. Shure, since they all exist in the ECM. Would the meaning of the sentence change, if you state: “The ECM is mainly composed of polysaccharides, proteins, and DNA”?

Line 11: You will find better formulations for that sentence!

Line 118: “… uptake of organic and inorganic molecules”. You summarize some organic examples in brackets. Do you have also some inorganic examples?

Line 134 – 137: You need a better formulation of that sentence.

Figure 2: You need a better explanation in terms of suspended cells vs biofilm cells. Are the cells in biofilms producing more autoinducer per cell than suspended cells do or is it just a matter of a suspension where the autoinducers have a greater mobility compared the ones in biofilms?

Line 192: Do you have an example that illustrate the harmlessness of the most biofilms? And for the adverse impact of microorganisms in food industry, too?

Line 208 – 223: Shorten the whole abstract.

Line 335: Shorten the sentence. I´m not sure, if I got its meaning.

Line 247 – 251: Please highlight literature for every point of your list.

Line 276: “First, steam heat treatment technology can operate in oxygen-free environment…”. What does exactly steam means in this context?

General remarks:

Generally, you need to underpin your statements with newer literature. In your outlook, I´m missing some techniques such as e-beam irradiation or plasma technologies, which are discussed in such a framework. Please give a broader view of possible future techniques. Discuss possible techniques against the background of its usage, e.g., are there any difference for various branches within the food industry.

Author Response

Response to Reviewer 2 Comments

This manuscript has been modified using Microsoft Word Track Changes. If you do not see any changes, click on the Review menu in Microsoft Word and select Final Showing Markup (or All Markup). See the red text in the full text of revised manuscript for details. Moreover, the language has been improved by some professional language editing.

Line 8 – 10: You are using the words microbial and extracellular a bit excessive. For instance, The ECM of biofilms is primarily composed of extracellular polysaccharides, extracellular proteins, and extracellular DNA”. Shure, since they all exist in the ECM. Would the meaning of the sentence change, if you state: “The ECM is mainly composed of polysaccharides, proteins, and DNA”?

Response:

Thank you very much for your correction. Due to the need to shorten the whole abstract, we have deleted this sentence (Line 9-11). As for the question you raised, we have found through literature review that there is indeed a problem with my expression. True: Bacterial cells are encapsulated in extracellular polymeric substances (EPS) composed of polysaccharides, proteins, nucleic acids, lipids, and other small molecules.

Line 11: You will find better formulations for that sentence!

Response:

Thanks for your suggestion. We have deleted this sentence in the revised abstract (Line 11-13).

Line 118: “… uptake of organic and inorganic molecules”. You summarize some organic examples in brackets. Do you have also some inorganic examples?

Response:

Thank you very much for your correction. We have added some inorganic examples (such as inorganic salt, water, etc.) to the original text (Line 118-120).

Line 134 – 137: You need a better formulation of that sentence.

Response:

Thank you very much for your correction. We have revised this sentence as below (Line 139-142).

In addition, bacteria usually integrate the information encoded in some QS automatic induction factors into the control of gene expression to achieve mutual communication between microorganisms (Papenfort & Bassler, 2016).

Figure 2: You need a better explanation in terms of suspended cells vs biofilm cells. Are the cells in biofilms producing more autoinducer per cell than suspended cells do or is it just a matter of a suspension where the autoinducers have a greater mobility compared the ones in biofilms?

Response:

Thank you very much for your comments and correction. We have completed the revision at the original text based on your comments. (Line 128-138 and Line 154-157)

Suspended cells have greater mobility than biofilm cells due to the low density of active space caused by the clustering of biofilm cells together. In addition, cells in biofilms produce more autoinducers than suspended cells because of the high number of cells in the biofilm, not because the cells are suspended. Autoinducer is a signaling molecule. These autoinducers are continuously produced by bacterial cells, so the level of autoinducers increases as the cell number increases. When autoinducers reach a minimum threshold range, these autoinducers are able to induce triggered signal transduction cascades that lead to multicellular responses in microbial populations. In other multicellular reactions, this mechanism can be involved in regulating biofilm formation, especially during the production of extracellular polysaccharides and the formation of channels or columnar structures. The formation of these structures ensures the transport of nutrients to cells in a biofilm community.

Line 192: Do you have an example that illustrate the harmlessness of the most biofilms? And for the adverse impact of microorganisms in food industry, too?

Response:

Thanks for your comments. In this article, we mean that the formation of biofilms is harmful in most cases, and then the article illustrates the harmful aspects of biofilm formation and some adverse effects on the food industry by giving examples (Line 204-219). Most microorganisms can adversely affect the food industry by forming biofilms. However, in the winemaking fermentation process, we need to promote the growth of Saccharomyces cerevisiae in large quantities. This is harmless for the food industry if beneficial microorganisms form biofilms during this process.

Line 208 – 223: Shorten the whole abstract.

Response:

Thank you very much for your suggestion. We have finished the modification according to your requirement (Line 8-24).

Line 335: Shorten the sentence. I´m not sure, if I got its meaning.

Response:

Thank you very much for your correction. We have revised this sentence by combining the preceding and following text (Line 346-349).

Line 247 – 251: Please highlight literature for every point of your list.

Response:

Thank you very much for your correction. The three main strategies for controlling harmful biofilms have been summarized in a literature (Yin et al., 2021), so the position of the reference literature has been revised in the original text (Line 259-263).

Line 276: “First, steam heat treatment technology can operate in oxygen-free environment…”. What does exactly steam means in this context?

Response:

Thanks for your comments. Steam heat treatment technology is very similar to autoclaving technology. It can operate in an anaerobic environment, which means that this technology has great lethality to aerobic bacteria. Heat treatment is equivalent to high temperature. Generally speaking, most bacteria cannot survive at high temperatures. In addition, steam can easily penetrate the cell's surface crevices or cracks. Therefore, steam heat treatment technology can effectively eliminate food-borne pathogens.

General remarks:

Generally, you need to underpin your statements with newer literature. In your outlook, I´m missing some techniques such as e-beam irradiation or plasma technologies, which are discussed in such a framework. Please give a broader view of possible future techniques. Discuss possible techniques against the background of its usage, e.g., are there any difference for various branches within the food industry.

Response:

Thank you very much for your comments and correction. Novel technologies for controlling or eliminating biofilms applied in the food industry are described in Table 2. Nonthermal atmospheric plasma technology is actually a kind of plasma technology, which is described in detail in Table 2. Through literature review, it is found that electron irradiation technology is a novel physical microbial inactivation technology, which has been classified in Table 2. In addition, we have added to the "Conclusions and future perspectives" section discussing the recent trend of the techniques used for the removal or prevention of biofilms in food items (Line 569-578).

Round 2

Reviewer 1 Report

The response from the author is satisfactory but still few comments need to address.

In the comparison with the previous article entitled “Biofilm formation and control strategies of foodborne pathogens: food safety perspectives”, this manuscript only changed the manner of discussing the content in the form of tables with a brief explanation. However, the novelty section remains the question. There are several articles, which discuss the almost similar concept, references have given below.  

For the recent trends of the techniques used for the removal of biofilms, instead of adding in the conclusion section, add a different section and discuss elaborately which will suit the title of the manuscript.

The section Summary of common foodborne pathogens and their implications for food safety – Possible mechanism discussion will not deviate the manuscript from the original concept it will add a more precise way of dealing with the possible way of combating the pathogen's virulence mechanism. In this regard, it will be good if the author discusses virulence mechanisms for the organisms, which have been discussed in the later section of the manuscript. Also the most specific way of combating the virulence of those particular organisms.

Bai, X.; Nakatsu, C.H.; Bhunia, A.K. Bacterial Biofilms and Their Implications in Pathogenesis and Food Safety. Foods 2021, 10, 2117, doi:https://doi.org/10.3390/foods10092117.